# In Vitro Antiviral Potential, Antioxidant, and Chemical Composition of Clove (*Syzygium aromaticum*) Essential Oil

**DOI:** 10.3390/molecules28062421

**Published:** 2023-03-07

**Authors:** Manal Jameel Kiki

**Affiliations:** The University of Jeddah, College of Science, Department of Biology, Jeddah 23218, Saudi Arabia; mjkiki@uj.edu.sa

**Keywords:** antiviral, antioxidant, cytotoxicity, clove, essential oil, *Syzigium aromaticum*

## Abstract

Viral infections are spread all around the world. Although there are available therapies, their safety and effectiveness are constrained by their adverse effects and drug resistance. Therefore, new natural antivirals have been used such as essential oils, which are natural products with promising biological activity. Accordingly, the present study aimed to identify the components of clove (*Syzygium aromaticum*) essential oil (EOCa) and verify its antioxidant and antiviral activity. The oil was analyzed using GC/MS, and the antioxidant capacity was evaluated as a function of the radical scavenging activity. A plaque reduction test was used to measure the antiviral activity against herpes simplex virus (HSV-1), hepatitis A virus (HAV), and an adenovirus. GC/MS analysis confirmed the presence of eugenol as the main component (76.78%). Moreover, EOCa had powerful antioxidant activity with an IC_50_ of 50 µg/mL. The highest antiviral potential was found against HAV, with a selectivity index (SI) of 14.46, while showing poor selectivity toward HSV-1 with an SI value of 1.44. However, no relevant effect was detected against the adenovirus. The antiviral activity against HAV revealed that its effect was not related to host cytotoxicity. The findings imply that EOCa can be utilized to treat diseases caused by infections and free radicals.

## 1. Introduction

Antiviral chemotherapy is an essential component of viral infection treatment. Many powerful medications have been developed during the last decade to combat viral infections, but their increased clinical usage or abuse has been linked to the rise in drug resistant strains [1]. However, potent antivirals for clinical application are only available against a few virus families. On the other hand, most of the world population still receives health care based on traditional medicine [2]. Plant phytochemicals have powerful antimicrobial and antiviral defense mechanisms. Therefore, secondary metabolites such as essential oils (EOs) generated from plant secondary metabolites have become increasingly prominent as possible sources of viral infection treatments in the last decade [1].

EOs are mixtures of aromatic volatile secondary metabolites that give plants their unique smell, taste, or both. They are created and reserved in secretory structures such as glands and may be found in various regions of plants as liquid droplets [3]. Despite having two or three major components at a concentration of 20–70%, EOs include a variety of chemicals such as aromatic hydrocarbons, terpenoids, terpenes, esters, acids, and alcohols. The percentage of each constituent gives the oil specific therapeutic effects. Many variables affect the chemical profile of EOs including their geographic location, the soil type, the season, the extraction method, and their storage. The great interest in EO research stems from its many medicinal and biological features as they are typically regarded as safe and have the potential to work in concert with other substances, which are both attractive characteristics for their usage as bioactive molecules [4].

The pharmacological characteristics of EOs have been investigated, and their various, antimicrobial, antiviral, antioxidant, anticancer, and anti-inflammatory properties have been described [5,6,7,8,9]. Therefore, due to their unique biological activities and physicochemical features, EOs have gained the attention of scientists in the past two decades [8]. There are over 3000 well-known EOs, but only 300 are regularly commercialized [4].

Cloves have been used for centuries in both traditional medicine and cooking, and their essential oil has been used in perfumery, traditional medicine, and food flavoring. Clove has traditionally been associated with enhancing the immune system and promoting resistance to illness. Although cloves have been generally used as a food spice in the West for hundreds of years, they are still used to treat a wide range of health problems as an anesthetic, antiseptic, antiviral, antifungal, and antimicrobial. Furthermore, clove essential oil has been used to treat burns, gum infections, digestive, respiratory, and other diseases. Previous research has shown additional important characteristics such as inflammatory, antimutagenic, and antioxidant capabilities [10].

Cloves are the dried flower buds of the evergreen tree *Syzgium aromaticum*, a member of the Myrtaceae family that grows in tropical areas. Cloves are now available as whole dried buds, as a powdered spice, and as an essential oil. Even though all forms possess comparable biomedical properties, oil is the most powerful. Whole cloves still have a medium level of potency because they contain a good amount of oil, but crushed cloves have the least potency because they lose a lot of their essential oil when they are crushed. However, the spice is high in minerals such as manganese, magnesium, iron, potassium, and selenium [10].

The essential oil of clove has been shown to have strong antimicrobial and antioxidant properties [11]. It has been proven that the strong antioxidant value of clove oil is connected to the phenolic compounds found in it [12,13]. The antioxidant power of the oil is attributed to its ability to minimize oxidation reactions and reduce the number of free radicals; therefore, it can be used in the formulation of medicines against the damage and disease caused by oxidative stress [14]. Moreover, it has been demonstrated that eugenol and eugenol acetate present in clove exhibit antioxidant properties [15]. The main constituent of clove essential oil (EOC) is eugenol, and it can be obtained via the distillation process [16]. The eugenol concentration of EOC has been observed to range between 45% and 90% in several investigations [11]. Several research groups have explored eugenol’s antiviral activity; it has shown antiviral action against Ebola, herpes simplex virus types 1 and 2 [17], and influenza A virus [18]. Recent research has indicated that eugenol derivatives might reduce the action of the West Nile virus, along with potential against flaviviruses such as zika virus, yellow fever, and dengue virus [19]. Eugenol has also been investigated as an effective inhibitor of the earliest stage of HIV-1 infection due its ability to limit viral replication. Similarly, eugenol may boost lymphocyte production; hence, eugenol’s potential for lymphocyte proliferation may be responsible for its anti-HIV-1 efficacy [20,21].

Currently, clove (*Syzygium aromaticum*) essential oil (EOCa) is widely available for various uses; therefore, the main purpose of the current study was to examine the chemical composition, antioxidant activity, and antiviral potential of the commercial clove essential oil (EOCa) against hepatitis A virus (HAV), herpes simplex virus (HSV), and an adenovirus.

## 2. Results and Discussion

### 2.1. GC/MS Analysis of Essential Oil

The sample of EOCa was analyzed using GC/MS, and the chemical constituents were identified and quantified, as illustrated in Table 1. The composition included eugenol (76.78%), β-caryophyllene (21.24%), α-copaene (1.16%), β-caryophyllene oxide (0.45%), and α-cubebene (0.28%). Altogether, five compounds were identified in the EOCa, representing 100% of the total compounds. The results showed that eugenol, β-caryophyllene, and α-copaene were the major represented molecules. The GC/MS results are in accordance with other studies reporting these compounds such as Jimoh et al. [22], who found that EOCa mainly contained 75.10% of eugenol and β-caryophyllene (5.2%). Another study [23] found that the main components of EOCa were eugenol (87%), eugenol acetate (8.01%), and (β-caryophyllene 3.56%).

In this study, eugenol (76.78%) represented the major component among the main compounds presents in EOCa. Similarly, a number of studies have found eugenol to be the main compound in EOCa, and it is considered as the molecule responsible for the numerous varieties of its activities [24,25]. On the other hand, the eugenol level can vary from one study to another, whereby other reports have revealed the presence of 82.6% eugenol in EOCa [26], a slightly higher percentage than in this study. Such high concentrations were also observed for eugenol (89.2%) [27]. In contrast, 59.5% eugenol was reported by Arslan et al. [28], whereas 59.9% eugenol was reported by Massodi et al. [29]. It has been suggested that the decline in eugenol could be related to the extraction system of essential oil [24]. In general, these oils have the same major composition, but the quantities of individual chemicals might vary depending on several variables such as the environment, genetics, and cultivation methods [9]. Moreover, it is recognized that the number of secondary compounds in essential oils are strongly affected by such factors, while essential oils from the same plant species might have distinct chemotypes [30].

### 2.2. Antioxidant Activity of Essential Oil

The inhibition percentages of EOCa were determined in an interval of 30 to 120 min using five concentrations within 50–800 µg/mL. The results illustrated in Table 2 reveal that EOCa is a powerful scavenger of DPPH radicals. It can be observed that its activity increased gradually with increasing concentration and time. The dose-dependent activity was observed for all concentrations; at 50 µg/mL, 47.6% activity was recorded at 30 min, increasing to 73.2% at 100 μg/mL. The antioxidant potential was found to be the maximum with 98.6% free-radical scavenging at 800 µg/mL. In general, all concentrations showed antioxidant activity at a substantial level.

The concentration required to scavenge 50% of the free radicals is called the IC_50_. A lower IC_50_ value indicates a higher antioxidant potential. The IC_50_ values were calculated using a graph plotting scavenging capability vs. concentration (Figure 1). According to the results, the IC_50_ value was 50 μg/mL in the DPPH free-radical-scavenging assays. The IC_50_ value for ascorbic acid was 10 μg/mL. These findings are consistent with Adefegha et al. [31], who showed that EOCa had an IC_50_ value of 43.7 g/mL according to the radical-scavenging activity.

The powerful antioxidant activity of clove may be due to its high phenolic component level (i.e., eugenol), which can protect cells from damage caused by free-radical oxidation (ROS). Some diseases such as Parkinson’s, Alzheimer’s, and cancer are associated with the presence of ROS compounds. The antioxidant properties of eugenol are caused by the hydroxyl group on the aromatic ring [32]. The phenolic compounds donate hydrogen atoms to free radicals and stabilize their structure, thereby blocking their oxidative process [21,33]. Furthermore, Dahham et al. [34] reported that a high level of antioxidant activity is shown by β-caryophyllene. Additionally, previous studies have shown that the antioxidant effect of EOCa is connected to the oil’s synergistic interaction of phenolic components and secondary metabolites [13]. Furthermore, several studies have revealed that most synthetic antioxidants cannot protect the body from ROS attacks if they are given in the inappropriate dose due to their toxic and carcinogenic effect [35]. Therefore, the use of antioxidant compounds derived from natural sources may aid in the removal of free radicals without affecting the body’s system to fight free radicals [13].

### 2.3. Evaluation of Antiviral Activity

The evaluation of viral activity was based on the IC_50_ value (50% infection concentration, i.e., the concentration of oil necessary to lower viral infection by 50%). The cytotoxicity was measured using the 50% cytotoxic concentration (CC_50_), which is the amount of oil necessary to inhibit 50% of the cells. The ratio of CC_50_ to IC_50_ was used to determine the antiviral selectivity index (SI), which serves as a measure of the therapeutic potential of the sample [36,37]. It is apparent from Table 3 that EOCa exhibited limited cytotoxicity to Vero76 cells, whereas the CC_50_ value was 14.21 ± 0.67 µg/mL. The plaque reduction assay was used to investigate the antiviral activities of essential oils against hepatitis A virus (HAV) (Figure 2), herpes simplex virus (HSV-1) (Figure 3), and adenovirus (Figure 4).

It appears from Table 4 that the antiviral efficiency of EOCa, as measured by the IC_50_ and SI, were different according to the types of viruses. However, this in vitro study showed that EOCa possessed a high antiviral activity against HAV, with an IC_50_ value of 0.73 ± 0.25 µg/mL and an SI of 14.46. In contrast, it had minimal antiviral efficacy against HSV-1, with an IC_50_ of 9.84 ± 0.68 µg/mL and an SI of 1.44, while there was no activity against the adenovirus. According to Hafidh et al. [1], when SI ≥10, it is usually assumed that the biological effect is not related to in vitro cytotoxicity. The findings showed that the best value of SI (14.46) was reported against HAV, demonstrating the antiviral efficiency of EOCa against HAV.

In this study, no viral inhibition was seen in the plaque reduction test in the adenovirus-infected cells. This finding agrees with Cermelli et al. [38], revealing that the adenovirus was unaffected by eucalyptus essential oil because it lacked a viral envelope. However, essential oils extracted from medicinal plants have been shown to be effective against several enveloped viruses. There have been several studies reporting that hepatitis B, herpes simplex, and human immunodeficiency virus multiplication capacities were inhibited by primeval treatment with essential oils, albeit by treating the cell with the oil prior to virus adsorption [39,40]. Although the results of this study showed that clove oil had a limited effect on the herpes virus, many studies have proven its effectiveness. For example, Benencia and Courrges [41] reported that eugenol, found in clove essential oil, slowed the progression of herpes virus-induced keratitis in mice. Eugenol was shown to be virucidal in the same investigation, and no cytotoxicity was seen at the doses used.

The available literature provides very limited information about the effect of different essential oils on HAV. Oils that have been previously studied include oregano, thyme, lemon, sweet orange, grapefruit, and rosemary [42,43]. However, there are no current studies on the effectiveness of EOCa against HAV.

Several studies have argued the efficacy of essential oils against viral infections; however, the specific mechanism behind this effect has not been completely elucidated. The results of previous studies revealed that the inhibitory effect of essential oils on the stages of viral infection cycle occurred during the processes of adsorption and penetration, not after the virus has already entered the cell [39,44,45].

On the other hand, eugenol, a volatile bioactive natural phenolic monoterpenoid that belongs to the phenylpropanoid family of natural compounds, was the major component of the clove oil examined in this research. It is widely known for its medicinal effects, which include antioxidant, antimicrobial, anti-inflammatory, analgesic, and anticancer activities. Moreover, β-caryophyllene-containing plants such as cloves might be useful in the selection of antivirals [46]. Kaur et al. [25] confirmed that a superior clove oil was distilled from buds that were rich in eugenol, β-caryophyllene, and eugenol acetate as the main compounds.

On the basis of the results discussed above, it can be suggested that the tested commercial EOCa used in this study can be considered as a high-quality essential oil, which can be used for therapeutic purposes. Previous studies showed that EOCa has a variety of medicinal applications, as a function of the presence of phenolic compounds, which are distinctive in that they possess both antioxidant and antimicrobial properties [11,13,47]. Each antioxidant has the capacity to function alone or in combination with other compounds to increase its antiviral action [48]. Furthermore, another study found that the powerful antioxidant activity connected to antimicrobial action is a consequence of the synergistic interaction among several substances [49]. Overall, the current study contributes to our knowledge about clove essential oil and highlights its potential antioxidant and antiviral activity. Further clinical research is required to validate these in vitro findings, evaluate its safety, and determine its future potential as an antiviral medication.

## 3. Material and Methods

### 3.1. Essential Oil

In this study, the clove (*Syzygium aromaticum*) essential oil (EOCa) sample was purchased from the local market of organic products in Jeddah city, Saudi Arabi.

### 3.2. GC/MS Analysis of Essential Oil

The chemical components of the essential oil were analyzed using an Agilent Technologies GC/MS system equipped with a mass spectrometer detector (5977A) and a gas chromatograph (7890B). The sample was diluted with hexane at a ratio of 1:19 (*v*/*v*). The GC was outfitted with an HP-5MS column (30 m × 0.25 mm, thickness 0.25 μm). For the analyses, helium was used as the carrier gas at a flow rate of 1.0 mL/min and at a 1:10 split ratio, with an injection volume of 1 µL and the following temperature schedule: 40/1; 4/1 to 150/6; 4/1 to 210/5 °C/min. The detector and injector were maintained at 220 and 280 °C, respectively. Electron ionization at 70 eV and in the range of *m*/*z* 40–550 were used to obtain the mass spectra. Mass spectra were collected using electron ionization at 70 eV, *m*/*z* 40–550, and a 3 min solvent delay. The EO components were specified by comparing the patterns of spectrum fragmentation to data from the NIST Mass Spectral and Wiley Libraries [50].

### 3.3. Antioxidant Activity of Essential Oil

The antioxidant activity of EOCa was evaluated using the technique of radical-scavenging capacity against 1,1-diphenyl-2-picrylhydrazyl (DPPH) according to the method of Park et al. [51]. The solution of each treatment was prepared by mixing 2 mL of the oil at different concentrations (50, 100, 200, 400, and 800 µg/mL) and 2 mL of DPPH (0.2 mM). The mixture was left in the dark at room temperature for 30, 60, 90, and 120 min. The absorbance was measured at 517 nm using a Perkin-Elmer 45 UV/Visible spectrophotometer. Ascorbic acid was used as the positive control, and all treatments were performed in triplicate. The following equation was used to determine the percentage inhibition: percentage inhibition (%) = absorbance of control − absorbance of sample/absorbance of control) × 100. The inhibition percentage was calculated, and the findings were reported as IC_50_ values (concentration that inhibits 50% of free radicals), which were calculated using a graph plotting the concentration versus scavenging capability [25].

### 3.4. Evaluation of Antiviral Activity

#### 3.4.1. Cells and Viral Culture

Vero cells were provided by the American Type Culture Collection (ATCC) and were grown in Dulbecco’s modified Eagle’s medium, which included 1% l-glutamine, HEPES buffer, 10% heat-inactivated fetal bovine serum, and 50 μg/mL gentamycin. All cells were kept at 37 °C and 5% CO_2_ in a humidified environment [52]. In confluent Vero cells, the cytopathogenic viral strains of herpes simplex virus (HSV-1), hepatitis A virus (HAV), and adenovirus were propagated and tested. The Spearman–Karber technique was used to enumerate the number of infectious viruses by estimating the 50% tissue culture infectious dose using eight wells per dilution and 20 µL of inoculum in each well [53].

#### 3.4.2. Cytotoxicity Assay

The cytotoxicity of EOCa toward Vero cells and the pertinent 50% cytotoxic concentration (CC_50)_ values were evaluated according to an established protocol, as reported by Mosman in a previous study [54]. The oil was tested using twofold serial dilutions within the 2.5–125 μg/mL concentration range. To assess the number of live cells, the optical density was determined at 590 nm using a microplate reader (SunRise, TECAN, Inc., Morrisville, NC, USA). The percentage viability was calculated as (ODt/ODc) × 100%, where ODt represents the mean optical density of the treated sample, and ODc represents the mean optical density of the untreated cells. The curve of the Vero cell survival was obtained by plotting the percentage of surviving cells against the oil concentration. The dose–response curves were established to obtain the (CC_50_) value using GraphPad Prism software.

#### 3.4.3. Antiviral Assay

The antiviral activity of EOCa was studied against the tested viruses using a plaque reduction assay according to Kaul et al. [55]. For the antiviral test, 0.1 mL of oil was mixed with 0.1 mL of 10^3^ virus suspension. The mixtures were left for 1 h at room temperature. The virus control was established by mixing the medium with 0.1 mL of 10^3^ virus suspension. Vero cell plates were washed and injected with 0.2 mL of either the test sample or the viral control. One well/plate was left uninoculated/untreated as a cell control. After washing, 2× MEM/agarose was applied to the cell monolayers. After solidification, the plate was inverted and incubated at 37 °C in 0.5% CO_2_.

After plaque formation, cells were fixed for 1 h at room temperature in a 10% formalin solution to inactivate the virus. After washing, 10% crystal violet solution was used to stain the cell monolayers for 10 min. Plaques were counted visually or using a stereomicroscope as a clear spot against a violet background of live cells. The number of plaque-forming units (PFU/mL) was used to describe the virus titer, according to the formula P × D (PFU)/mL = V, where D is the virus dilution, P is the number of plaques, and V is the virus dilution. All tests employed untreated virus-infected cells as the controls. The antiviral effect was defined by a 50% decrease in the plaque count formed by the treated virus compared to the untreated virus (IC_50_).

### 3.5. Statistical Analysis

All experimental outcomes were tabulated as the mean ± standard deviation of three independent replicates. The values of CC_50_ and IC_50_ were determined on the basis of graphs of the dose–response curve at each concentration using GraphPad Prism 5 software.

## 4. Conclusions

Clove (*Syzygium aromaticum*) essential oil (EOCa) is characterized by numerous therapeutic and biological properties. In this study, it exhibited powerful antioxidant capacity due to the phenolic compounds. Eugenol was identified as the main component in EOCa. Antiviral activity was tested against herpes simplex virus (HSV-1), hepatitis A virus (HAV), and an adenovirus. The highest antiviral potential was found against HAV, with poor selectivity toward HSV-1. However, no relevant effect was detected against the adenovirus. It is worth mentioning that the available literature provides very limited information about the effect of different essential oils on the HAV. The oils that have been previously studied include oregano, thyme, lemon, sweet orange, grapefruit, and rosemary. Moreover, there are no current studies on the effectiveness of clove essential oil against HAV. Crucially, this in vitro study found clove essential oil to have high antiviral activity against HAV. In conclusion, the antiviral selectivity index against HAV indicated no relevance between the activity and the cytotoxicity of the host cells. Consequently, EOCa can be used as a natural food supplement, antioxidant, and antimicrobial. The current findings are supported by the well-documented biological and antimicrobial efficacy of clove essential oil in conventional medicine. In addition, future studies on the current topics are suggested to explore the potential effect of EOCa and other essential oils on the human microbiome and immunity.

## Figures and Tables

**Figure 1 molecules-28-02421-f001:**
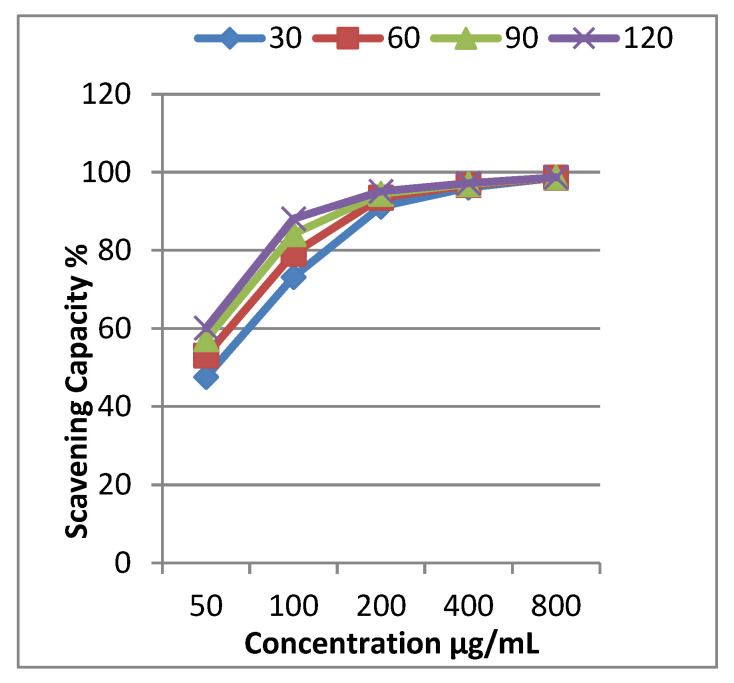
Free-radical-scavenging activity of clove essential oil and IC_50_ values for scavenging 50% of the free radicals DPPH (IC_50_ = 50 μg/mL).

**Figure 2 molecules-28-02421-f002:**
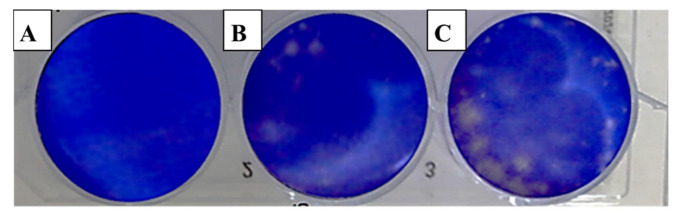
Effect of clove essential oil on plaque formation by hepatitis A virus (HAV) in cell culture using the plaque reduction assay: (**A**) Vero cells (control); (**B**) Vero cells infected with HAV and treated with oil; (**C**) Vero cells infected with HAV (infection control).

**Figure 3 molecules-28-02421-f003:**
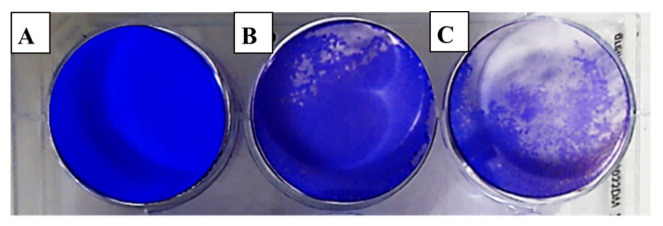
Effect of clove essential oil on plaque formation by the herpes simplex virus (HSV-1) in cell culture using the plaque reduction assay: (**A**) Vero cells (control); (**B**) Vero cells infected with HSV-1 and treated with oil; (**C**) Vero cells infected with HSV-1 (infection control).

**Figure 4 molecules-28-02421-f004:**
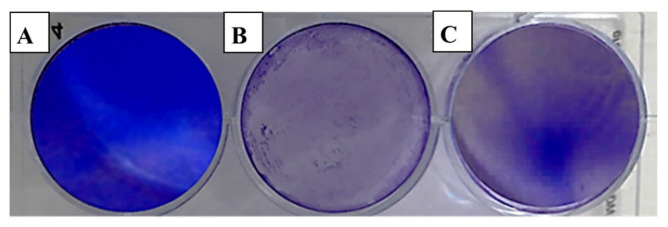
Effect of clove essential oil on plaque formation by the adenovirus in cell culture using the plaque reduction assay: (**A**) Vero cells (control); (**B**) Vero cells infected with adenovirus and treated with oil; (**C**) Vero cells infected with the adenovirus (infection control).

**Table 1 molecules-28-02421-t001:** Chemical composition of clove (*Syzygium aromaticum*) essential oil.

Name	Formula	Retention Time	Area	Area Sum %
α-Cubebene	C_15_H_24_	21.704	914,030.44	0.28
Eugenol	C_10_H_12_O_2_	22.264	253,303,146.8	76.78
α-Copaene	C_15_H_24_	22.567	3,839,412.04	1.16
β-Caryophyllene	C_15_H_24_	23.971	70,203,334.66	21.24
β-Caryophyllene oxide	C_15_H_24_O	28.838	1,793,977.82	0.54

**Table 2 molecules-28-02421-t002:** Antioxidant activity of clove (*Syzygium aromaticum*) essential oil against 2,2-diphenyl-1-picrylhydrazyl hydroxyl radicals.

Concentrations (µg/mL)	Inhibition (%) ± SD			
	30	60	90	120
50	47.6 ± 0.62	53.1 ± 0.28	57.3 ± 0.24	60.1 ± 0.12
100	73.2 ± 0.22	79.2 ± 0.14	84.2 ± 0.13	88.1 ± 0.23
200	91.2 ± 0.21	93.5 ± 0.32	94.5 ± 0.05	95.1 ± 0.12
400	96.1 ± 0.11	96.7 ± 0.25	97.0 ± 0.17	97.2 ± 0.03
800	98.6 ± 0. 15	98.6 ± 0.01	98.7 ± 0.21	98.7 ± 0.25

The outcomes are presented as the mean ± standard deviation (SD).

**Table 3 molecules-28-02421-t003:** Cytotoxic effect of clove essential oil against Vero cells (50% cell cytotoxic concentration CC_50_ = 14.21 ± 0.67 µg/mL).

Sample Concentration (µg/mL)	Viability (%)	Cytotoxicity (%) ± SD
0.25	97.82	2.18 ± 0.46
0.5	95.66	4.34 ± 0.32
1	89.74	10.26 ± 0.52
2	80.28	19.72 ± 0.46
3.9	72.49	27.51 ± 1.03
7.8	59.17	40.83 ± 1.79
15.6	48.02	51.98 ± 1.34
31.25	34.83	65.17 ± 1.29
62.5	20.95	79.05 ± 0.63
125	8.04	91.96 ± 0.28

The outcomes are presented as the mean ± standard deviation (SD).

**Table 4 molecules-28-02421-t004:** The antiviral efficiency of clove essential oil against the tested viruses.

Virus	Cytotoxic Effects CC_50_ µg/mL ± SD	Antiviral Effects IC_50_ µg/mL ± SD	Selectivity Index (SI)
Hepatitis A virus (HAV)	14.21 ± 0.63	0.73 ± 0.25	14.46
Herpes simplex virus (HSV-1)	14.21 ± 0.63	9.84 ± 0.68	1.44
Adenovirus	14.21 ± 0.63	NA	NA

CC_50_: concentration causing 50% cell cytotoxic effects; IC_50_: concentration causing 50% antiviral effects; SI = CC_50_/IC_50_; NA: no activity; SD: standard deviation.

## Data Availability

Not applicable.

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
