# Peer review of "In Vitro Antiviral Potential, Antioxidant, and Chemical Composition of Clove (Syzygium aromaticum) Essential Oil"

_molecules, 2023, doi:10.3390/molecules28062421_

Round 1

Reviewer 1 Report (Previous Reviewer 1)

Dear Author,

the manuscript is now available for publication.

Kind regards,

Reviewer 2 Report (Previous Reviewer 2)

No comments

This manuscript is a resubmission of an earlier submission. The following is a list of the peer review reports and author responses from that submission.

Round 1

Reviewer 1 Report

Comments to the Author

The manuscript entitled “In Vitro Antiviral Potential, Antioxidant, and Chemical Composition of Clove (Syzygium aromaticum) Essential Oil” has been reviewed. The topic is quite innovative, interesting and scientifically well structured.

The MS is acceptable after minor revisions.

Introduction.

the introduction should contain a brief description of the essential oil concept.

A brief botanical description of Syzygium aromaticum could also enrich the MS

Conclusion

the conclusions could be improved by identifying some new research trends that could be proposed from the results obtained in the MS.

Reviewer 2 Report

·      Please give details of the plant material origin, and the date of harvesting and essential oil extraction.

·      Please add the GC analysis conditions.

·      GC-MS analysis is insufficient for the identification of known substance. Usually, two orthogonal criteria are required, most frequently these are GC-MS plus retention indices calculation and comparison with those of literature. Co-injection with available sample of authentic compound for confirmation of assignment made is also recommended.

·      Mass spectroscopy detector do not consider response factor, how author quantify the essential oil composition? Quantification needs to be done by using a method which considers “response factor”.

·      It’s surprising for me that clove essential oil is composed of only five molecules!

·      Author reported that “The powerful antioxidant activity of clove may be due to its high phenolic component level, i.e., eugenol and β-caryophyllene, …”. Please correct the sentence β-caryophyllene is not a phenolic compound.

·      Ensure that all the conclusions are listed.

Round 2

Reviewer 2 Report

Relative percentage of essential oil composition should be determined using a detector that considers response factor, like flame ionization detector. Unlike, mass spectroscopy detector does not consider response factor. For future research, I suggest for authors to consult the following papers regarding the preparation, identification, and quantification of volatile compounds: https://doi.org/10.1002/ffj.3433 and https://doi.org/10.1002/ffj.3445